# Assisting Standing Balance Recovery for Parkinson’s Patients with a Lower-Extremity Exoskeleton Robot

**DOI:** 10.3390/s24237498

**Published:** 2024-11-24

**Authors:** Chi-Shiuan Lee, Lo-Ping Yu, Si-Huei Lee, Yi-Chia Chen, Chun-Ta Chen

**Affiliations:** 1Department of Mechatronic Engineering, National Taiwan Normal University, 162, Section 1, He-Ping East Rd., Taipei 106, Taiwan; s920365@gmail.com; 2Attending Physician of Department of Physical Medicine and Rehabilitation, Taipei Veterans General Hospital, 201, Section 2, Shipai Rd., Taipei 106, Taiwan; yuloping@yahoo.com.tw (L.-P.Y.); leesihuei@gmail.com (S.-H.L.); 3Department of Electrical Engineering, National Kaohsiung Normal University, 62, Shenzhong Rd., Kaohsiung 824, Taiwan; lisa9219710@gmail.com

**Keywords:** exoskeleton robot, lower-extremity, standing balance assistance, zero moment point, fuzzy sliding mode control, Parkinson’s disease

## Abstract

Parkinson’s disease (PD) is a neurodegenerative disorder and always results in balance loss. Although studies in lower-extremity exoskeleton robots are ample, applications with a lower-extremity exoskeleton robot for PD patients are still challenging. This paper aims to develop an effective assistive control for PD patients with a lower-extremity exoskeleton robot to maintain standing balance while being subjected to external disturbances. When an external force is applied to participants to force them to lose balance, the hip strategy for balance recovery based on the zero moment point (ZMP) metrics is used to generate a reference trajectory of the hip joint, and then, a model-free linear extended state observer (LESO)-based fuzzy sliding mode control (FSMC) is synthesized to regulate the human body to recover balance. Balance recovery trials for healthy individuals and PD patients with and without exoskeleton assistance were conducted to evaluate the performance of the proposed exoskeleton robot and balance recovery strategy. Our experiments demonstrated the potential effectiveness of the proposed exoskeleton robot and controller for standing balance recovery control in PD patients.

## 1. Introduction

Balance is fundamental for daily living activities. Balance and postural control for falling prevention are accomplished by implementing sensorimotor control strategies, and thus, stability is ensured [1]. Age-related dysfunctions may lead to balance loss and result in a dramatic increase in falling. Moreover, people with mobility deficits as a result of neuromusculoskeletal disorders or impaired sensorimotors such as stroke, Parkinson’s syndrome, spinal cord injury and other diseases have poor balance ability and are frequently reported to lose balance and fall.

Balance control ability can be improved by repeatedly delivering a physical disturbance to train individuals [2,3]. Although the benefit of improving balance in standing and gait has been proven, there are still risks of falling in non-well-trained individuals due to severe impairment or disorder in neuromotors. This leads to the required use of crutches to maintain fall-preventing balance. However, crutches limit a user’s ability to execute activities of daily living without stress [4,5,6,7].

Recent studies on lower-extremity exoskeleton robots have shown their potential applications to power assistance or rehabilitation exercise and could become assistive devices for physiological activities [8]. The wearable mechanical structure type of device whose links are strapped to human limbs can fundamentally improve the individuals’ physical abilities by providing active assistance [9,10,11,12]. And thus, wearable lower-extremity exoskeleton robots could also provide an opportunity to execute various motor tasks. While exoskeleton robots have shown reliability in locomotion assistance, balance support is also in serious demand for reducing the risk of falling. However, improving the safety and performance of exoskeletons with novel functionalities still remains an ongoing challenge.

Currently, standing and walking balance control are the crucial functions in promoting the usage of exoskeleton robots [13]. Some metrics are provided to evaluate balance. Commonly, the human body’s center of mass (COM) [14,15] and the center of pressure (COP) on the foot are selected as the balance metrics [16,17]. Balance is maintained if the COM or COP is kept over the feet, or inside the support boundary that is circumscribed by both the standing feet. The other effective solution to judge the balance of human locomotion is the stability criterion of the zero moment point (ZMP) [18], in which ZMP is defined as a point on the ground about which the total moments of the foot–ground contacts are zero [19]. Specifically, the extrapolated center of mass (XCoM) obtained by vertically projecting the position of CoM to the ground in the direction of its velocity can be formulated as the balance control during walking [20].

Depending on the balance metrics, different control techniques were employed to maintain the stability of upright posture for balance. If the defined metrics move away from their support boundary, a human will be regarded off-balance, and typically a fall tendency happens. Then, an exoskeleton robot should immediately take a corrective movement strategy to react to disturbances to return to a balance state. The required postural strategies for balance control and balance recovery vary depending on perturbation type, intensity, reaction ability and motor constraints [21].

For balance recovery following disturbances during walking, Leestma et al. [22] presented a bilateral bang–bang controller for an autonomous robotic hip exoskeleton that modulates step width and length in all cardinal and ordinal directions during steady state and perturbed walking. However, the step modulation capability is influenced by swing leg kinematics and perturbation context. A series elastic actuator-driven compliant hip exoskeleton to assist with movement and maintain balance was developed in [23], in which an extrapolated center of mass (XCOM) concept for walking stability was employed for the balance control design. A compliant guidance force is produced to react to perturbations in balance. In [24], inspired by human balance responses, a cooperative ankle-exoskeleton control strategy was presented to assist in balance recovery after unexpected disturbances while walking. The results show that the active controller was able to cooperate with the able-bodied participants in counteracting perturbations. Qin et al. [25] developed a self-coordinated velocity vector double-layer controller with balance-guiding ability for a lower-limb rehabilitation exoskeleton robot. However, the optimality-based balance trajectory planning is time-consumed on the implementation of walking balance.

With respect to the standing balance, the balance control strategies used within exoskeleton robots, including an ankle strategy, a hip strategy, combined strategy and a stepping strategy, are typically from observations of balance disturbance experiments on unimpaired individuals. Horak and Nashner [26] used the ankle and hip strategies for balance recovery following external perturbations. The inverted pendulum control of posture is involved for the ankle strategy; the hip strategy consists of anti-phase ankle plantarflexion and hip flexion rotations. The study investigated only the selected pattern of activation in a given trial in advance. Instead of torque control strategies, a bioinspired approach was adopted by Fasola et al. [27] to identify and then to implement the postural strategies for standing balance on a complete spinal cord injury (SCI) user with a lower-limb exoskeleton robot. Standing quietly, resisting external perturbations and lifting barbells of increasing weight with the exoskeleton worn by an SCI pilot were tested for the perturbation-resisting ability. The balance control is achieved by learning in a passive exoskeleton and then porting onto an active exoskeleton with equivalent mobility. Not including torque control in joints or load cells in the feet can keep the hardware minimal. An assist-as-needed control method that could aid both ankle and hip strategies through joint angle-dependent stiffness and stability limits was presented in [28]. But the limits of stability of each individual were determined using a Wii platform during the initialization phase. In [29], Emmens et al. subsequently introduced exoskeleton control of quiet standing and fixed-support balance recovery using classical proportional-derivative (PD) control of whole-body COM. Subjects adapt their ankle torques to the torques provided by the ankle exoskeleton to maintain balance. An experimental study on balance recovery control with a lower-limb exoskeleton robot was presented in [30]. Participants subjected to a forward force perturbation during standing were forced to step forward with the right leg by the exoskeleton assistance for balance recovery. Balancing performance was compared with and without exoskeleton assistance for the investigation of the influence of the exoskeletons’ control assistance. In the study by Zhang et al. [31], a method based on the capture point concept was proposed to enhance the balance restoration capabilities of the exoskeleton under significant interference conditions. However, this study did not address the standing issue. Li et al. [32] presented a model predictive control framework for standing balance recovery in lower-limb exoskeleton robots. The orbit energy metrics were proposed to trigger appropriate strategy. However, the torque inputs to regulate the capture point within the base of support must be optimized for the model predictive control. Furthermore, a larger external push may need alternative balance strategy. In [33], Orhan et al. introduced a stepping strategy for a push recovery framework for collaborative human–exoskeleton systems. The stepping strategy was formulated as an online optimization problem to determine the optimal step position and duration so that the recovery of balance was allowed under severe external disturbances.

In daily activities, standing for freehand tasks or environment interactions is very common. Moreover, as most of the exoskeleton robots used for standing balance trials are assessed with respect to able-bodied participants, very few investigations have been reported for the impaired. Among them, over 4 million people worldwide are affected by PD, which is a chronic, progressive and long-term neurodegenerative disorder [34], characterized by the motor symptoms of tremor, stiffness, bradykinesia and postural instability [35]. The postural instability weakens the ability to maintain standing balance and increases the risk of falling during everyday activities. Therefore, the goal of this paper is to develop a balance recovery controller for PD patients with a lower-extremity exoskeleton robot to react to unexpected disturbances or self-generated perturbations during standing. Our main hypothesis is that a forward thrust is applied to participants, and the assistant hip joint torques from the exoskeleton robot are enough to counteract the disturbance. Consequently, a hip strategy for balance recovery based on the ZMP metrics is used to generate a reference trajectory of the hip joints, and then, a model-free LESO-based FSMC is synthesized to reflexively regulate the human body to recover balance. As compared with the referred publications using the model-based controller for balance control, more system parameters are required for better performance, and these parameters should be distinguished for different subjects. However, the proposed LESO-based FSMC can implement the balance recovery control more effectively without the need of the dynamic model. In addition, most studies on balance recovery always assume an off-balance while experiencing a pushing disturbance. However, our proposed methodology is reliable in detecting the off-balance tendency through the insole force sensors that detect the variations in COP. If the COP is beyond the specified stability boundary, the imbalance is notified, and then, the assistance for standing balance recovery reflexively starts up until the COP returns to the specified stability region, while also being capable of reducing participants’ effort to recover balance. Moreover, to the best knowledge of the authors, few investigations into the standing balance recovery for PD patients with an exoskeleton robot assistance have been conducted.

## 2. Lower-Extremity Exoskeleton Robot Design

Lower-extremity exoskeletons are used to provide assistance to a user. It indicates that the power is transmitted to the lower extremity through the straps of the exoskeleton robot. As a consequence, the exoskeleton robot is better able to move in compliance with the kinematics of the human’s lower extremity. Therefore, the lower-extremity exoskeleton robot is designed in line with the ergonomic principles such that it can be kinematically consistent with the lower extremity of humans [8,36].

### Mechanical Structure and Control Circuit

The designed lower-extremity exoskeleton robot has an active flexion/extension at each hip and knee joint and a passive flexion/extension at each ankle joint to comply with the movement of lower limbs as shown in Figure 1a. At each hip joint, a DC brushless motor (MAXON EC 60 flat, Switzerland) connected with the 1:100 harmonic drive is used for the flexion/extension actuation. The MAXON-ECI52 brushless motors connected to 1:46 reducers (IDP gear 42046, Taiwan) actuate the calf links fixed to the end of the thigh links at the knee joints. As shown in Figure 1b, a human wears the proposed lower-extremity exoskeleton.

The control circuit for the exoskeleton robot is shown in Figure 2, in which the NI sbRIO-9632 (USA) consisting of a real-time controller, reconfigurable IO modules (RIO), FPGA modules and an Ethernet expansion chassis is employed for real-time control. The four channels of the NI 9234 buffer, condition and sample the input signals through a 24-bit Delta-Sigma ADC. The output analog signals are enabled by NI 9263 through four channels that have the specification of ±10 V, 16-bit, 100 kS/s. The controller design and feedback measurements are implemented using the Labview (v8.5) system.

## 3. Standing Balance Recovery and Stability

One of the standing balance strategies is to keep ZMP within the support polygon due to its perfect working in the balance control. The ZMP is related to the acceleration of COM. When a pilot is subjected to an external disturbance, the ZMP will be away from the initial balance position. ZMP formulation can be used to generate the reference acceleration of COM for balance recovery. The reference trajectory of COM is then transformed to the required reference joint trajectory of the exoskeleton robot, and then, a controller is designed according to the joint angle differences. The outputs from the controller serve as the compensated angles of the joints so that the actual ZMP will be used to return to the balance position.

### 3.1. Zero Moment Point

If the ZMP is located within the polygon formed by the foot–ground contact boundary, the human can maintain stable walking or maintain a standing balance. The x-coordinate of ZMP on the ground of the sagittal plane is determined as [19]
(1)$$x_{ZMP}=x_{G}-\frac{z_{G}{\ddot{x}}_{G}}{{\ddot{z}}_{G}+g},$$
in which $r_{G}={\left[\begin{array}{cc} x_{G} & z_{G} \end{array}\right]}^{T}$ is the position of COM on the sagittal plane, and *g* is the gravitational acceleration. From the expression, the ZMP is related to the position and acceleration of the human’s COM that can be defined at the pelvis [37].

### 3.2. Balance Recovery Strategy

When a standing subject is subjected to a push disturbance, the ZMP will deviate from the initial position until reaching the presumed stability boundary. The lower-extremity exoskeleton robot will then be enabled to apply torques at the hip joints to assist the standing balance according to the hip strategy, while the knee and ankle joints are still controlled by the subject. The proposed balance recovery strategy for the exoskeleton robot is to design the reference accelerations of the COM, and hence, the corresponding position of the COM is controlled. By regulating the position of COM, the deviated ZMP may be moved back to the initial stable position.

Currently, the balance recovery is considered on the sagittal x-z plane as shown in Figure 3. The reference position xrefZMP defines a desired ZMP position at which a disturbed human needs to regain the balance stability. From Equation (1), the reference acceleration of COM is designated as
(2)x¨Gref=z¨refG+gzG(xG−xrefZMP),

In Equation (2), the reference acceleration of COM in the *x* direction for the current posture is also related to the reference z-directional acceleration z¨Gref of COM. During the recovery process, z¨Gref implying the reference recovery acceleration in the z direction can be designated in a virtual spring-damping model as
(3)z¨Gref=kdz˙Gd−z˙G+kpzGd−zG,
in which ${\dot{z}}_{G}^{d}$ and $z_{G}^{d}$ are, respectively, the desired z-directional velocity and position. In general, they are designated as the initial standing state. $k_{d}$ and $k_{p}$ are constants dominating the reference recovery acceleration. If the z-directional velocity and position of COM approaches the desired state, the reference acceleration z¨Gref will be close to zero.

Taking the integral on the reference acceleration, the reference velocity r˙G=x˙Grefz˙GrefTref of COM is obtained. Furthermore, the corresponding reference joint velocities θ˙ref=θ˙hθ˙kθ˙aTref for hips, knees and ankles are determined by the Jacobian relation and pseudo inverse as
(4)θ˙ref=J†r˙Gref,
in which ${J^{†}=(J^{T}J)}^{-1}J^{T}$ defines the pseudo inverse of the Jacobian ***J***.

## 4. Controller Design

In this study, we aim to address the main problems affecting the assistance of standing balance recovery with an exoskeleton robot: each user’s different lower limb parameters and unknown exerted torques, as well as efficiently reactive control. To this end, an LESO-FSMC is proposed through a model-free estimator and controller for unknown dynamics compensation and balance control reinforcement.

### 4.1. LESO Design

The exoskeleton robot is a human–machine collaboration system, the wear’s lower limbs and the exoskeleton parameters are coupled to balance recovery dynamics. The standing dynamic model can be expressed as
(5)$$M\left(\theta \right)\ddot{\theta }+V\left(\theta ,\dot{\theta }\right)+G\left(\theta \right)=\tau_{r}+\tau_{h},$$
in which $\theta ={\left[\begin{array}{ccc} \theta_{1} & \theta_{2} & \theta_{3} \end{array}\right]}^{T}={\left[\begin{array}{ccc} \theta_{h} & \theta_{k} & \theta_{a} \end{array}\right]}^{T}$, $\theta_{h}$, $\theta_{k}$, $\theta_{a}$ are the hip, knee and ankle joint angles; $M\left(\theta \right)$, $V\left(\theta ,\dot{\theta }\right)$, and $G\left(\theta \right)$ denote the inertia matrix, the Coriolis and centrifugal matrix, and the gravitational vector, respectively. $\tau_{r}={\left[\begin{array}{ccc} \tau_{r1} & \tau_{r2} & \tau_{r3} \end{array}\right]}^{T}={\left[\begin{array}{ccc} \tau_{rh} & \tau_{rk} & \tau_{ra} \end{array}\right]}^{T}={\left[\begin{array}{ccc} \tau_{rh} & 0 & 0 \end{array}\right]}^{T}$ are the driving torque vectors. Note that knee and ankle joints of the exoskeleton are passive. $\tau_{h}={\left[\begin{array}{ccc} \tau_{h1} & \tau_{r2} & \tau_{r3} \end{array}\right]}^{T}={\left[\begin{array}{ccc} \tau_{rh} & \tau_{rk} & \tau_{ra} \end{array}\right]}^{T}$ are the torques that are, respectively, exerted at the corresponding joints by the human.

To estimate the unknown terms $\tau_{h}$ in real time, a model-free disturbance observer LESO was employed without a detailed mathematic model [38]. The methodology extends to another state including dynamic uncertainties and human-exerted torques, and then, the combined disturbance is estimated using the pole placement method.

Equation (5) can be expressed by taking an inverse on the inertia matrix and introducing a diagonal control gain matrix $b_{0}=diag[b_{01},\;b_{01},\;b_{01}]$ as
(6)$$\ddot{\theta }=f+b_{0}u,$$
where $f=M^{-1}\left(\tau_{h}-V-G)\right)+M^{-1}u-b_{0}u$ accounts for the combined effects of uncertain dynamics and unknown human’s exerted torques on joint angular acceleration, and $u=\tau_{r}$.

It is seen that Equation (6) is uncoupled, so LESO for each joint can be designed independently. Without loss of generality, by defining $x_{1}=\theta_{h}$, ${\dot{x}}_{1}=x_{2}$, ${\dot{x}}_{2}=x_{3}+b_{0h}u_{rh}$ and ${\dot{x}}_{3}={\dot{f}}_{h}$, an augmented state space for the hip joint is expressed as
(7)$$\left\{\begin{array}{c} \dot{x}=Ax+Bu+Eh\\ y=Cx \end{array}\right.,$$
in which $A=\left[\begin{array}{ccc} 0 & 1 & 0\\ 0 & 0 & 1\\ 0 & 0 & 0 \end{array}\right]$, $B={\left[\begin{array}{ccc} 0 & b_{0h} & 0 \end{array}\right]}^{T}$, $C=\left[\begin{array}{ccc} 1 & 0 & 0 \end{array}\right]$, $E={\left[\begin{array}{ccc} 0 & 0 & 1 \end{array}\right]}^{T}$, $h={\dot{f}}_{h}$ being part of the jerk and physically bounded, and $x_{3}=f_{h}$ is the added augmented state.

According to the state space expression for dynamic Equation (7), the state observer estimating the state of (7) can be designed as [34]
(8)$$\left\{\begin{array}{c} \dot{z}=Az+Bu+L\left(y-\hat{y}\right)\\ \hat{y}=Cz \end{array}\right.,$$
in which $z=[\begin{array}{ccc} z_{1} & z_{2} & z_{3} \end{array}{]}^{T}$, being the estimation of the system state $x=[\begin{array}{ccc} x_{1} & x_{2} & x_{3} \end{array}{]}^{T}$, is the state vector of the observer, ***L*** = ${\left[\begin{array}{ccc} \beta_{1} & \beta_{2} & \beta_{3} \end{array}\right]}^{T}$ is the observer gain vector, and $\hat{y}$ is the estimate of the system output *y*.

In (8), $z_{3}$ is the estimation of *f*. Moreover, the tracking errors of the observer are defined as ***e*** = ***x*** − ***z***, the error dynamics are then derived by Equations (7) and (8) as
(9)$$\dot{e}=\left(A-LC\right)e+Eh=A_{e}e+d,$$
in which $A_{e}=A-LC=\left[\begin{array}{ccc} -\beta_{1} & 1 & 0\\ -\beta_{2} & 0 & 1\\ -\beta_{3} & 0 & 0 \end{array}\right]$.

The observer gains are determined such that the characteristic polynomial $f\left(\lambda \right)=\left|\lambda I-A_{e}\right|$ is Hurwitz-type and can be parameterized by the bandwidth. In this regard, ***L*** = ${\left[\begin{array}{ccc} \beta_{1} & \beta_{2} & \beta_{3} \end{array}\right]}^{T}={\left[\begin{array}{ccc} 3\omega_{0} & {{3\omega }_{0}}^{2} & {\omega_{0}}^{3} \end{array}\right]}^{T}$.

### 4.2. LESO-Based FSMC Design

Compared to traditional LADRC that is synthesized by the linear feedback control of the estimated system state, sliding mode control (SMC) is an effective technique relative to the parametric uncertainties and external disturbances and has been successfully applied to many nonlinear systems due to the characteristics of simple control and easy implementation.

Note that only hip joints are assisted by the exoskeleton robot for balance recovery; thus, a controller is designed for the hip joints. The other joints, i.e., knee and ankle, are controlled by the wearer. To ensure an accurate tracking of the reference motion of the hip joint, θhipref, integrated using the hip velocity extracted from Equation (4), i.e., θhipref=θhip+θ˙hipref∆t, a LESO-based FSMC is developed.

Let us define a time-varying sliding surface:(10)s=ce+e˙,e=θhipref−θhip.

A sliding mode controller comprises the nominal control $u_{eq}$ that is determined by making the derivative of the sliding surface zero and the fuzzy type of reaching control $u_{r}$ for the system disturbances. Overall, the control input takes the form
(11)ut=ueqt+urt=θ¨hipref−fh^+ce˙/b0+(αFSMCs,s˙)/b0,
in which the positive coefficients α can be defined by the following stability analysis. The fuzzy function *FSMC* accounting for the uncertainties and eliminating the chattering maps two normalized inputs *s*(*t*), and $\dot{s}$(*t*) to the linguistic output. The Mamdani-inferred rules with seven fuzzy partitions, NB (Negative Big), NM (Negative Medium), NS (Negative Small), ZO (Zero), PS (Positive Small), PM (Positive Medium) and PB (Positive Big), are used for the fuzzy inference. The membership functions of input and output linguistic variables are defined in Figure 4. As proposed in [39], the product inference with singleton fuzzification and centroid defuzzification methods is used for the fuzzy implications. The fuzzy function is normalized as $\left|FSMC\left(s,\dot{s}\right)\;\right|\leq 1$ and has been set as (s)($FSMC\left(s,\dot{s}\right))\leq \left|s\right|$.

A Lyapunov candidate *V*(*t*) is chosen as
(12)$$V=\frac{1}{2}s^{2},$$

The stability is analyzed by differentiating Equation (12) as
(13)$$\dot{V}=s\dot{s}=s\left(-f_{h}+\hat{f_{h}}+\alpha FSMC\right)\;=s∆f+\alpha sFSMC\;\leq \left|s\right|\left|∆f\right|-\alpha \left|s\right|\;=\left|s\right|(\left|∆f\right|-\alpha ),$$
in which $∆f=-f+\hat{f}$ is the observed disturbance error.

During the dynamic variation, the parameters α must be chosen such that $\alpha >\left|∆f\right|$ to let $\dot{V}<0$ for e≠0 to remain on the sliding mode surface. Moreover, as time t→∞, from Equation (5) with $A_{e}$ being Hurwitz, $\left|∆f\right|$ is bounded, and the system is thus guaranteed stable.

### 4.3. Assistance to Standing Balance Control

Relying on the proposed standing balance recovery strategy and LESO-FSMC, the block diagram for assistance implementation is presented as in Figure 5. Because COP, defined as a point at which the resultant reaction force from the ground acts on the foot, can be measured by the force sensors mounted to the insole to express the ZMP-like point on the ground in real time [40], the COP is thus employed as the real-time standing stability measurement and performance evaluation.

When a human is pushed by a disturbance force, the insole with 180 distributed force sensors (Lisole Smart Insole, Taiwan) detects the variations in COP. If the COP is beyond the specified stability boundary, the unbalance is notified, and then, the reference accelerations and velocities of the COM are calculated. The reference trajectories of the COM are further mapped to the reference hip joint angles that the exoskeleton robot needs to follow to recover the balance soon after the human is pushed by a thrust. To control the balance recovery effectively, the LESO-FSMC is used for trajectory tracking at the hip joint. As such, the lower-extremity exoskeleton drives the human body to return to the balance posture.

Finally, the proposed balance recovery control strategy is benchmarked against [32] for the performance of control strategy, and the model predictive control framework needs to optimize the torque inputs. However, optimization methodology is time-consuming. Moreover, the orbit energy metrics in [32] are not reliable to trigger an appropriate strategy in comparison with our insole force sensors-based COP measurement.

## 5. Evaluation on Assistance of Standing Balance Recovery

In the section, the performance of the proposed ZMP-based hip strategy controller for the lower-extremity exoskeleton robot on standing balance recovery is investigated. For healthy subjects, the performance for different controllers, stability boundary, and thrusts are explored. A compared Fuzzy-proportional-derivative controller is taken as
(14)$$u\left(t\right)=\left(k_{p}e+k_{d}\dot{e}\right)+\beta FSMC\left(s,\dot{s}\right),$$
in which the tracking error e=θhipref−θhip, $k_{p}$ and $k_{d}$ are the respective proportional and derivative gain, and the positive coefficient β controls the system stability.

Moreover, the assistant effects on standing balance recovery are assessed for PD patients with different rated stages with and without an exoskeleton robot. During the trial process, an emergency push button will be prompted to cut off the power supply if the monitor code detects an excessive driving current. In addition, if the COP has returned inside the presumed stability region and the corresponding hip joint angle is less than 3°, a balance recovery will be regarded to have been achieved, and the exoskeleton robot will not provide a sustained assistance.

The required parameters for the reference acceleration and the proposed LESO- FSMC are taken as follows: ($k_{p}$,$\;k_{d}$) = (9, 3) in the reference z acceleration, $\omega_{0}$= 80/s in the LESO, $b_{0}=$12/kg·m^2^, *c* = 45, α = 60 for the FSMC.

### 5.1. Assistance Control for Healthy Subject

As shown in Figure 6, a healthy subject with 1.7 m height/87 kg weight wore the exoskeleton robot to implement standing balance recovery while being subjected to 7.4 kg of push disturbance on his back. The stability region was set between 6 cm and 17 cm from the heel. If the COP was beyond this region, the exoskeleton robot would execute balance assistance rapidly. The stability variations during the recovery process are presented by the COP positions measured by the insole sensors.

#### 5.1.1. Assistance Performance on Balance Recovery

In this experiment, the balance recovery evaluation for the subject with and without the exoskeleton was conducted. The stability region was assigned 6~17 cm from the heel. Figure 7a–d present the deviation positions of the COM and COP, as well as the hip joint angle variations. Without external assistance, the subject’s maximum deviation positions of COM and COP are, respectively, 10.58 cm and 21 cm, and the hip joint angle has the maximum 18.69°. In addition, it takes 1.5 s to recover the body to the balance posture.

However, with balance recovery assistance, the exoskeleton rapidly responds to a push and then assists the human body to return to the balance. The proposed LESO-FSMC results in the maximal deviation positions of 8.57 cm in COM, 19.35 cm in COP, 4.2° in the maximum hip joint angle, and 1 sec in balance recovery as compared to the Fuzzy-PD with the corresponding deviation positions of 9.86 cm for COM, 20.13 cm for COP, 16.96° for the maximal hip joint angle, and 1.3 s for balance recovery. Also, during the balance recovery process, Fuzzy–proportional-derivative always presents the human–exoskeleton antagonism. However, the observer-based FSMC identifies the exotic disturbance including the human applied torque so that the antagonism can be counteracted. It is seen that the LESO-FSMC has better performance on balance recovery.

#### 5.1.2. Assistance Evaluation for Different Stability Regions

In the subsection, the balance recovery with exoskeleton robot assistance using LESO-FSMC was investigated for different stability regions, in which a stability region was set as 6~17 cm from the heel, and the other one was a smaller region of 7~15 cm. After the subject was pushed by an 8.9 kg force, the COM deviation positions, COP deviation positions and the hip joint angles are shown in Figure 8a–d. It is seen that the tight safety region (7–15 cm) can respond to the push disturbance more rapidly, and then provide the assistance such that COM and COP present the smaller deviations, as well as smaller hip joint angles before the subject returns to a balance position.

#### 5.1.3. Balance Recovery for Different Thrusts

Balance recovery for three different magnitude of thrusts, i.e., 7–8 kg, 8–9 kg and larger than 9 kg, was explored. A larger thrust will lead to larger COM, COP deviation positions and hip joint angles as shown in Figure 9a–d. Specifically, the thrust of larger than 9 kg would make the COP position shift to the foot margin. Moreover, as the body is pulled back, and again goes beyond the stability region, a larger chattering is generated by the exoskeleton robot.

### 5.2. Balance Assistance for PD Patients

The investigation on standing balance recovery for PD patients wearing the proposed exoskeleton robot was conducted at the Attending Physician of Department of Physical Medicine and Rehabilitation, Taipei Veterans General Hospital. Because of the tentative investigation on standing balance recovery for PD patients, subjects with mild PD symptoms were selected. Therefore, two PD patients who, respectively, were rated with the Hoehn and Yahr scale stage 1 and stage 2 on standing balance activities (DT) attended the clinical trials as shown in Figure 10 [40]. All experimental protocols were approved by the TMU-Joint Institutional Review Board. Following preceding procedures, the PD patients stood with the proposed exoskeleton robot being worn. A thrust was applied to the patients’ back. If the subjects identified a balance loss, then the exoskeleton robot would be triggered to assist the balance recovery until the patients recovered to the equilibrium state. The COM variations, the COP variations and hip joint angles were measured and calculated to investigate the effectiveness for PD patients with/without lower-extremity exoskeleton assistance.

The COM variations in the X and z directions, the COP variations and hip joint angles for the PD patient in stage 1 are depicted in Figure 11a–c, in which the thrusts are 3.37 kg without assistance and 3.79 kg with assistance. With the assistance of the exoskeleton robot, it is seen that the maximum COM displacements in the x and z direction are reduced. Also, in the comparisons of COP deviation displacement, the COP has a smaller deviation of 18.4 cm with the assistance of exoskeleton robot, while the COP deviation is 21.2 cm without an assistance. The hip joint angle variations for the balance restoring process are presented in Figure 11d. It is shown that the exoskeleton robot can assist in decreasing the maximum hip joint angle 17.2° that implies the tilt angle of upper body, and just takes 1.5 s to return to the upright posture rapidly. Compared to PD’s self-balancing without any assistance, the maximum hip joint angle 21.2° and the spending time 2.03 s to recover balance are higher than that with exoskeleton assistance.

The balance assistance to the PD patient of stage 2 was demonstrated. The patient had an apparent shuffling gait of small steps and bradykinesia. The variations in COM and COP, hip joint angles and actuated torques with/without assistance are displayed in Figure 12a–d. The patient was subjected to a 3.33 kg thrust without assistance and had a maximum hip joint angle of 15.05° and the balance recovery time of 2.25 s. However, the maximum hip joint angle and the balance recovery time individually decreased to 9.16° and 1.1 s for the subject with assistance under a thrust of 3.33 kg. This trial also demonstrated the rapid and available response for the PD patient with a higher stage.

Different stages of PDs reflect the affected severity of the nervous system and the parts of the body controlled by the nerves and, thus, how they affect the balance control. Based on the balance assistance trials, the performance of the balance recovery for different stages of PDs is summarized in Table 1. Because PD always causes stiffness and slowing of movement, it was found that the maximum hip joint angle for the subject in te stage 2 was smaller than the one in stage 1. Without assistance, the subject in stage 2 spent 2.25 sec. to recover balance, longer than the time taken, 2.0 s, for the subject in stage 1 as a result of worse balance control for patients in high-rated stages. However, with the assistance of the exoskeleton robot, the balance recovering time was reduced substantially for the subject in stage 2 than stage 1 since the ability to perform the movements was decreased for more serious PD patients. Consequently, the human–exoskeleton antagonism was reduced more.

## 6. Conclusions

This study bridges the application gap of an exoskeleton robot to PD patients for standing balance recovery. In comparison with most existing balance assistance algorithms based on model-based control, we investigated the effects of a lower-extremity exoskeleton robot with the ZMP-metrics-based model-free LESO-FSMC controller on the balance recovery of normal subjects and PD patients receiving external perturbations during standing. With the insole sensors, the presented approach is capable of responding to an imminent standing balance loss rapidly and, then, counteracting the disturbances to assist users to recover balance promptly.

More notably, our findings for the trials on PD patients with exoskeleton robot assistance show the potential of the lower-extremity exoskeleton robot with the proposed controller to assist the PD patients’ standing balance recovery promptly to respond to external disturbances, reducing the user’s effort. The results obtained with participants show promise for exoskeleton usage to assist people with motor impartments in improving standing balance control. Further research will focus on walking balance control and other balance strategies in response to other types of perturbations, using our exoskeleton robot as the test platform.

## Figures and Tables

**Figure 1 sensors-24-07498-f001:**
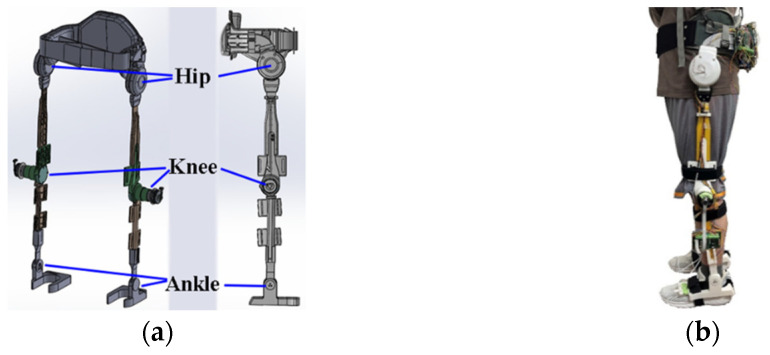
(**a**) Design of robotic hip–knee exoskeleton. (**b**) Building and wearing of proposed robotic knee–hip exoskeleton.

**Figure 2 sensors-24-07498-f002:**
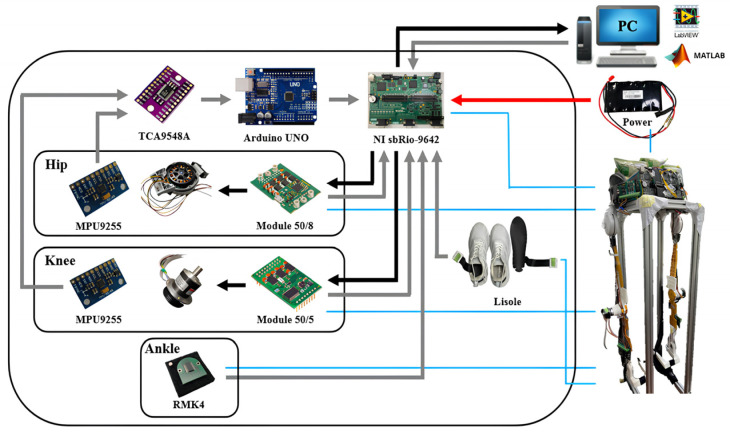
Control circuit and peripherals.

**Figure 3 sensors-24-07498-f003:**
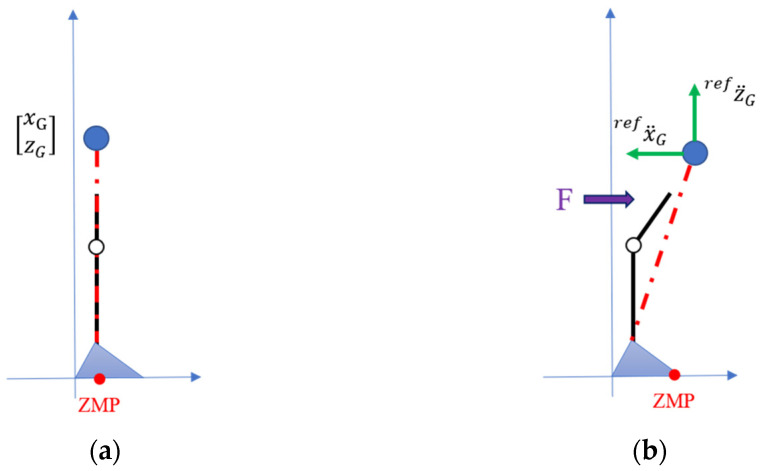
(**a**) Initial standing posture; (**b**) ZMP-based hip strategy for balance recovery.

**Figure 4 sensors-24-07498-f004:**
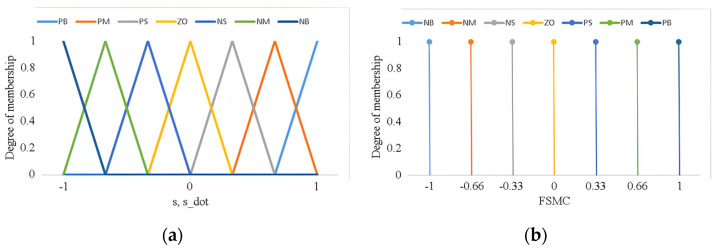
Assigned membership function of fuzzy sets for (**a**) input variables (s,s_dot), and (**b**) output function FSMC.

**Figure 5 sensors-24-07498-f005:**
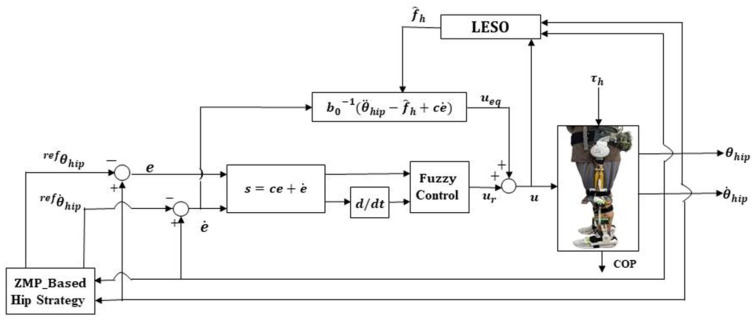
Control structure of LESO-based FSMC for balance recovery assistance.

**Figure 6 sensors-24-07498-f006:**
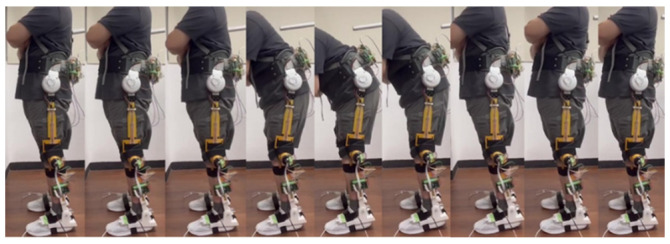
Experiment on standing balance recovery while being subjected to push disturbance.

**Figure 7 sensors-24-07498-f007:**
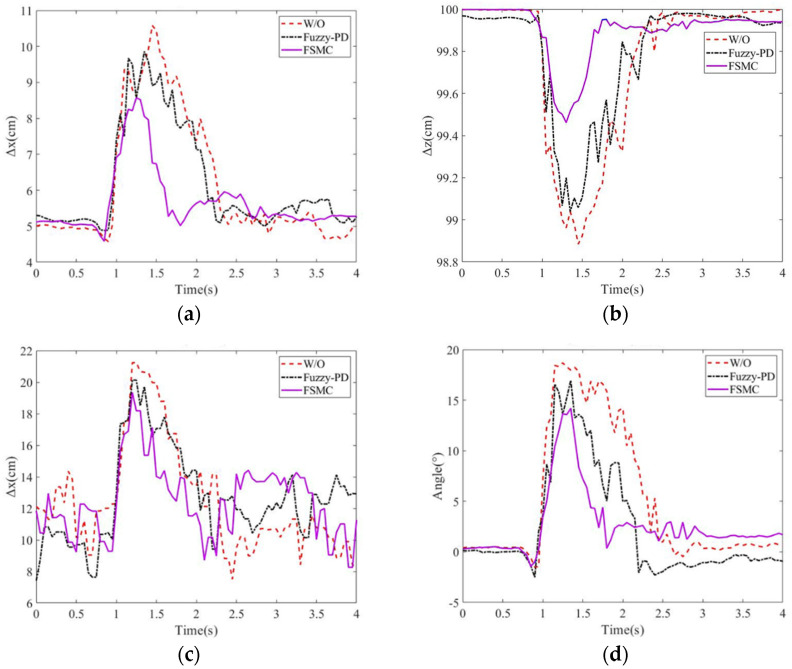
Position variations in (**a**) COM along x direction; (**b**) COM along z direction; (**c**) COP; (**d**) hip joint angles for different controls.

**Figure 8 sensors-24-07498-f008:**
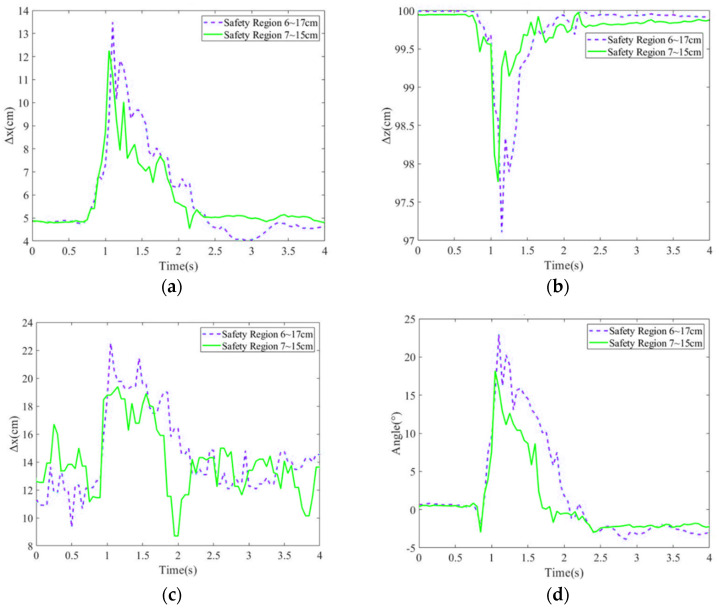
Position variations in (**a**) COM along x direction; (**b**) COM along z direction; (**c**) COP; (**d**) hip joint angles for different stability regions.

**Figure 9 sensors-24-07498-f009:**
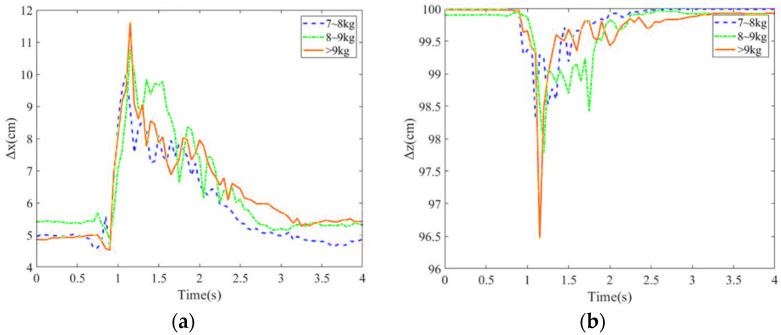
Position variations in (**a**) COM along x direction; (**b**) COM along z direction; (**c**) COP; (**d**) hip joint angles for different magnitude of thrusts.

**Figure 10 sensors-24-07498-f010:**
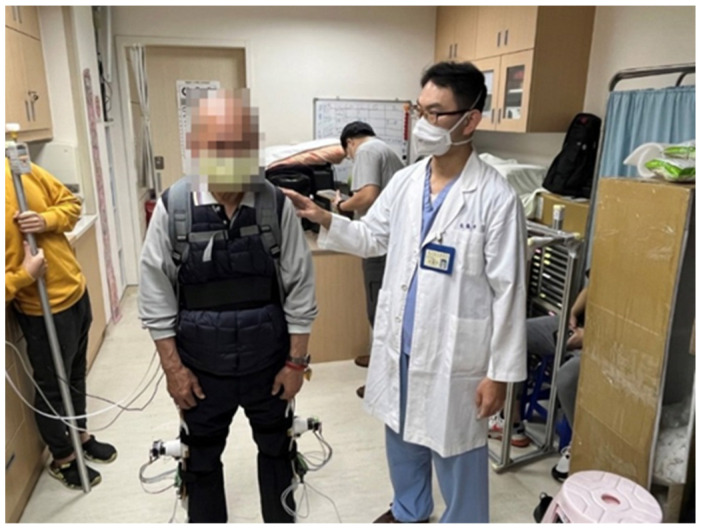
Clinical trials on standing balance recovery for PD patients with proposed exoskeleton robot.

**Figure 11 sensors-24-07498-f011:**
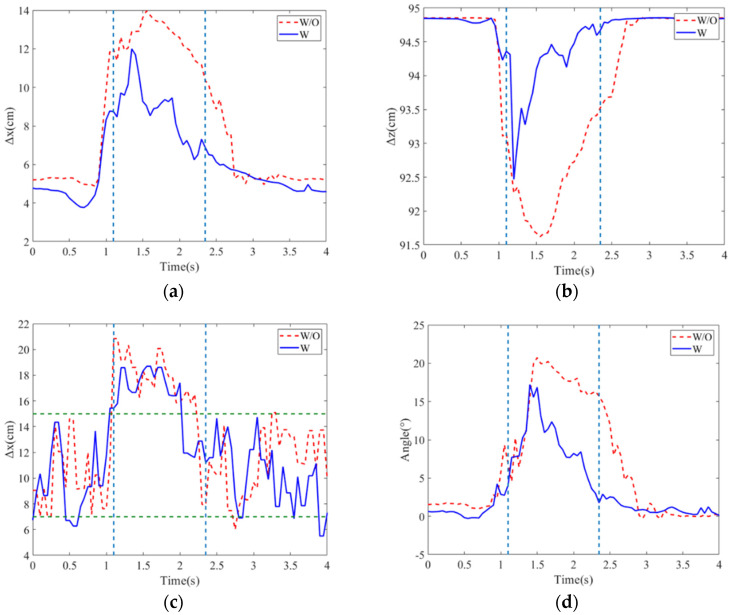
Position variations in (**a**) COM along x direction; (**b**) COM along z direction; (**c**) COP; (**d**) hip joint angles for PD patient of stage 1 with/without exoskeleton robot. The dashed vertical lines are on/off time of assistance, and the dashed horizontal lines are stability region.

**Figure 12 sensors-24-07498-f012:**
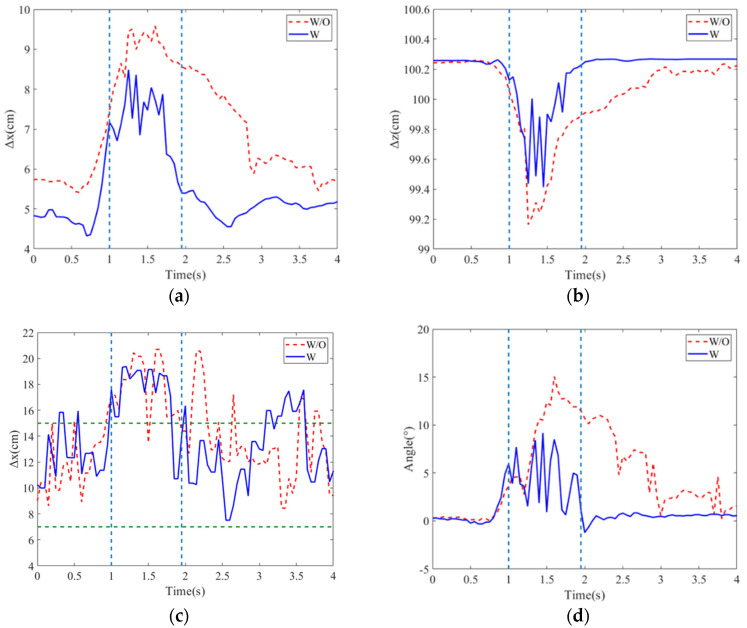
Position variations in (**a**) COM along x direction; (**b**) COM along z direction; (**c**) COP; (**d**) hip joint angles for PD patient of stage 2 with/without exoskeleton robot. The dashed vertical lines are on/off time of assistance, and the dashed horizontal lines are stability region.

**Table 1 sensors-24-07498-t001:** Balance recovery comparisons for PD patients of different stages.

	Stage 1		Stage 2	
	WO (3.37 kg)	W (3.79 kg)	WO (3.35 kg)	W (4.11 kg)
COMX (cm)	13.97	11.99	9.51	8.45
COMZ (cm)	91.71	92.42	99.15	99.4
COP (cm)	21.2	18.4	20.85	19.02
$\mathrm{Max}.\;\theta_{hip\;}({}^{\circ})$	20.72	17.2	15.05	9.16
BRT (s) ^1^	2	1.5	2.25	1.1

^1^ BRT = balance recovery time.

## Data Availability

The data that support the findings of this research are available from the corresponding author, [C.T. Chen], upon reasonable request.

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
