# Peer review of "Assisting Standing Balance Recovery for Parkinson’s Patients with a Lower-Extremity Exoskeleton Robot"

_sensors, 2024, doi:10.3390/s24237498_

Round 1
Reviewer 1 Report
Comments and Suggestions for Authors
This paper developed an effective assistive control for PD patients with a lower extremity exoskeleton robot. Its design was introduced, control strategy was explained and its effectiveness was verified in experiments. Overall, this paper demonstrates clear novelty and the contents are clearly written overall. The reviewer can recommend for publication of the following comments can be addressed.
1. Too many abbreviations are used in this paper. To improve the readability of this paper, table of acronyms should be added.
2. Page 2, line 96, what does ‘SCI’ stand for?
3. Page 3, the key novelties and contributions of this paper should be more clearly summarized in the end of the introduction section.
4. Figure 2 and 4 are in poor resolutions in the current manuscript file, please improve the quality.
5. Page 9, line 335, ‘com’ should be ‘cm’.
6. The reviewer suggests the authors to ensure a consistent format in presenting the numbers. For example, a space is needed between ’20.13’ and ‘cm’, i.e. ‘20.13 cm’.
7. This paper includes human experiments, the authors are recommended to provide ethics approval for conducting these human experiments.
8. Please benchmark the performance of the control strategy proposed in Section 4.3 against the state-of-the-art control strategies for the similar exoskeleton robots.
9. Please include the units for the parameters for the reference acceleration and the proposed LESO-FSMC in line 311 to 312.
10. The term Fuzzy-PD can be confused with The Parkinson's disease (PD) in this paper despite that PD is a standard control term, please rename this term.
Author Response
Response Note to the Reviewers
We appreciate your valuable comments. Thank you very much for encouraging us to resubmit a revised version of this work. We appreciate and agree with your valuable suggestions. The authors substantially revised it based on reviewers’ comments. Please note that revisions are made in red words since improvements have been made to the submitted paper.
Reviewer 1
This paper developed an effective assistive control for PD patients with a lower extremity exoskeleton robot. Its design was introduced, control strategy was explained and its effectiveness was verified in experiments. Overall, this paper demonstrates clear novelty and the contents are clearly written overall. The reviewer can recommend for publication of the following comments can be addressed.
- Too many abbreviations are used in this paper. To improve the readability of this paper, table of acronyms should be added.
- Page 2, line 96, what does ‘SCI’ stand for?
- Page 3, the key novelties and contributions of this paper should be more clearly summarized in the end of the introduction section.
- Figure 2 and 4 are in poor resolutions in the current manuscript file, please improve the quality.
- Page 9, line 335, ‘com’ should be ‘cm’.
- The reviewer suggests the authors to ensure a consistent format in presenting the numbers. For example, a space is needed between ’20.13’ and ‘cm’, i.e. ‘20.13 cm’.
- This paper includes human experiments, the authors are recommended to provide ethics approval for conducting these human experiments.
- Please benchmark the performance of the control strategy proposed in Section 4.3 against the state-of-the-art control strategies for the similar exoskeleton robots.
- Please include the units for the parameters for the reference acceleration and the proposed LESO-FSMC in line 311 to 312.
- The term Fuzzy-PD can be confused with The Parkinson's disease (PD) in this paper despite that PD is a standard control term, please rename this term.
Author response:
- Thank you for your positive comments. A list of acronyms has been added in the appendix.
- SCI is the acronym of spinal cord injury. In the revision, the complete words were added.
- The key novelties and contributions of this paper are clearly summarized in the end of the introduction section in the revision. As compared with the referred publications using the model-based controller for balance control, more system parameters are required for better performance, and these parameters should be distinguished for different subjects. However, the proposed LESO-based FSMC can implement the balance recovery control more effectively without the need of the dynamic model. In addition, most studies on balance recovery always presume an off-balance while experiencing a pushing disturbance. However, our proposed methodology is reliable in detecting the off-balance tendency through the insole force sensors that detect the variations of COP. If the COP is beyond the specified stability boundary, the unbalance is notified, and then the assistance for standing balance recovery reflexively starts up until the COP returns to the specified stability region, meanwhile also capable of reducing participants’ effort to recover balance. Moreover, to the best knowledge of the authors, few investigations to the standing balance recovery for PD patients with an exoskeleton robot assistance were conducted. Please see the end of the introduction section.
- The quality of Figure 2 and 4 has been improved in the revision. Please see the Figure 2 and 4.
- In the revision, ‘com’ has been corrected as ‘cm’. Please see the Page 9, line
- We appreciate the reviewer’s suggestions, the authors have checked the paper to ensure a consistent format in presenting the numbers.
- The ethics approval for conducting the human experiments has been added. Please see the Institutional Review Board Statement at page 15.
- The proposed balance recovery control strategy is benchmarked against [32] for the performance of control strategy, the model predictive control framework needs to optimize the torque inputs. However, optimization methodology is time-consuming. Moreover, the orbit energy metrics in [32] is not reliable to trigger appropriate strategy in comparison with our insole force sensors-based COP measurement. Please see section 4-3.
- The units for the parameters for the reference acceleration and the proposed LESO-FSMC are included for and , and some parameters are gains with no units. Please see the page 10.
- The term Fuzzy-PD has been renamed as Fuzzy-proportional-derivative. Please see page 9.
Reviewer 2 Report
Comments and Suggestions for Authors
Comments and Suggestions for Authors
The paper presented an effective assistive control for PD patients with a lower-extremity exoskeleton robot to maintain standing balance while being subjected to external disturbances. However, there are some major issues with this paper that need to be addressed in the revision:
a) The article referred to the application of exoskeleton robots in facilitating the restoration of standing balance; however, the descriptions regarding the specific working mechanism, technical principles of the exoskeleton robots, and how they achieve balance assistance are relatively cursory. It is suggested that the author add descriptions of the relevant technical details.
b) Although this article introduces the clinical trial of standing balance restoration in patients with Parkinson's disease, the details and specific methods of the experiment are slightly brief. It is suggested that the author add more descriptions of the experimental design, including the selection criteria of subjects, experimental procedures, data collection and analysis methods, etc., so that readers can understand and reproduce the experimental results more comprehensively.
Some specific comments requiring attention:
1. The figures need to be better handled to make them more readable, e.g. changing the color combination, line width, fonts and markers.
2. The parameters appearing in the formula 14 should be indicated;
3. Ensure that Figure 11 and Figure 12 are in the same format as other legends
4. Figures should be placed in the appropriate position in the text and as centered as possible or arranged in the format required by the journal.
5. The font size should be consistent in the figures.
6. There are numerous unnecessary spaces in between the words throughout the paper.
7. I suggest that more references should be added for a deeper understanding of the existing work.
Comments on the Quality of English Language
The manuscript in its present form has some simple English usage errors. Grammar errors as well as the formatting of text should be checked and improved throughout the article.
Author Response
Response Note to the Reviewers
We appreciate your valuable comments. Thank you very much for encouraging us to resubmit a revised version of this work. We appreciate and agree with your valuable suggestions. The authors substantially revised it based on reviewers’ comments. Please note that revisions are made in red words since improvements have been made to the submitted paper.
Reviewer 2
The paper presented an effective assistive control for PD patients with a lower-extremity exoskeleton robot to maintain standing balance while being subjected to external disturbances. However, there are some major issues with this paper that need to be addressed in the revision:
a. The article referred to the application of exoskeleton robots in facilitating the restoration of standing balance; however, the descriptions regarding the specific working mechanism, technical principles of the exoskeleton robots, and how they achieve balance assistance are relatively cursory. It is suggested that the author add descriptions of the relevant technical details.
Author response:
We have added the descriptions of the relevant technical details as the following: When a human is pushed by a disturbance force, the insole with 180 distributed force sensors (Lisole Smart Insole) detects the variations of COP. If the COP is beyond the specified stability boundary, the unbalance is notified, and then the reference accelerations and velocities of the COM are calculated. The reference trajectories of the COM are further mapped to the reference hip joint angles that the exoskeleton robot needs to follow to recover the balance soon after the human is pushed by trust. To control the balance recovery effectively, the proposed controller is used for the trajectory tracking at the hip joint. As such, the lower-extremity exoskeleton drives the human body to return to the balance posture. Please see the section 4.3.
b. Although this article introduces the clinical trial of standing balance restoration in patients with Parkinson's disease, the details and specific methods of the experiment are slightly brief. It is suggested that the author add more descriptions of the experimental design, including the selection criteria of subjects, experimental procedures, data collection and analysis methods, etc., so that readers can understand and reproduce the experimental results more comprehensively.
Author response:
The investigation on standing balance recovery for PD patients wearing the proposed exoskeleton robot was conducted. Because of the tentative investigation on standing balance recovery for PD patients, the subjects with the mild PD symptoms were selected. Therefore, two PD patients who respectively were rated with the Hoehn and Yahr scale stage 1 and stage 2 on standing balance activities attended the clinical trials as shown in Fig. 10. The PD patients stood with the proposed exoskeleton robot being worn. A trust was applied to the patients’ back. The insole with 180 distributed force sensors (Lisole Smart Insole) detects the variations of COP. If the COP is beyond the specified stability boundary, the unbalance is notified, and then the reference accelerations and velocities of the COM are calculated. If the subjects are identified a balance loss, then the exoskeleton robot will be triggered to assist the balance recovery until the patients recover to the equilibrium state. The COM variations, the COP variations and hip joint angles were measured and calculated to investigate the effectiveness for PD patients with/without lower-extremity exoskeleton assistance. Please see the section 5.2.
Some specific comments requiring attention:
- The figures need to be better handled to make them more readable, e.g. changing the color combination, line width, fonts and markers.
Author response:
In the revision, the color combinations of all the figures were changed to make more readable.
- The parameters appearing in the formula 14 should be indicated;
Author response:
The parameters appearing in the formula 14 have been defined and explained. Please see the page 9.
- Ensure that Figure 11 and Figure 12 are in the same format as other legends.
Author response:
Figure 11 and Figure 12 present the comparison of the performance on standing balance recovery for the PD patients at different stages with/without an exoskeleton robot. Therefore, the legends show the w and w/o formats.
- Figures should be placed in the appropriate position in the text and as centered as possible or arranged in the format required by the journal.
Author response:
In this revision, the figures are placed as centered as possible.
- The font size should be consistent in the figures.
Author response:
In this revision, the font sizes are in consistence in all the figures.
- There are numerous unnecessary spaces in between the words throughout the paper.
Author response:
We check throughout the paper and try to delete unnecessary spaces in between the words
- I suggest that more references should be added for a deeper understanding of the existing work.
Author response:
More references have been added in the revision. Please see the introduction and the added references 28, 30-33.